# MAU: A Motion-Aware Unit for Video Prediction and Beyond

**Zheng Chang**
University of Chinese Academy of Sciences
Institute of Computing Technology, Chinese Academy of Sciences
`changzheng18@mails.ucas.ac.cn`

**Xinfeng Zhang**
School of Computer Science and Technology,
University of Chinese Academy of Sciences
`xfzhang@ucas.ac.cn`

**Shanshe Wang** [*]
Institute of Digital Media,
Peking University
`sswang@pku.edu.cn`

**Siwei Ma**
Institute of Digital Media,
Information Technology R&D Innovation Center,
Peking University
`swma@pku.edu.cn`

**Yan Ye**
Alibaba Group
`yan.ye@alibaba-inc.com`

**Xinguang Xiang**
School of Computer Science and Engineering,
Nanjing University of Science and Technology
`xgxiang@njust.edu.cn`

**Wen Gao**
Institute of Digital Media,
Peking University
University of Chinese Academy of Sciences
`wgao@pku.edu.cn`

## Abstract

Accurately predicting inter-frame motion information plays a key role in video prediction tasks. In this paper, we propose a Motion-Aware Unit (MAU) to capture reliable inter-frame motion information by broadening the temporal receptive field of the predictive units. The MAU consists of two modules, the attention module and the fusion module. The attention module aims to learn an attention map based on the correlations between the current spatial state and the historical spatial states. Based on the learned attention map, the historical temporal states are aggregated to an augmented motion information (AMI). In this way, the predictive unit can perceive more temporal dynamics from a wider receptive field. Then, the fusion module is utilized to further aggregate the augmented motion information (AMI) and current appearance information (current spatial state) to the final predicted frame. The computation load of MAU is relatively low, and the proposed unit can be easily applied to other predictive models. Moreover, an information recalling scheme is employed into the encoders and decoders to help preserve the visual details of the predictions. We evaluate the MAU on both video prediction and early action recognition tasks. Experimental results show that the MAU outperforms the state-of-the-art methods on both tasks.

---

[*]Corresponding author: Shanshe Wang.

35th Conference on Neural Information Processing Systems (NeurIPS 2021).

# 1 Introduction

Video prediction is a representative task in video predictive learning area, which aims to predict the unknown future on the basis of the limited knowledge and has been applied in a wide range of research areas, such as robotic control [1], video interpolation [2], autonomous driving [3], motion planning [4] and so on. However, compared with images, videos are more complex due to the time-varying motion information and predicting reliable motion information has always been a significant but challenging problem for video prediction tasks. Fortunately, deep learning technologies have shown their great power in learning meaningful features for multimedia data and have achieved great success in computer vision and natural language processing tasks. Motivated by this, learning-based methods have been applied for video prediction in recent years.

Recurrent neural networks (RNNs) are first applied to learn video representations due to their unique advantages in modeling sequential data [5]. Then the Long Short-Term Memory (LSTM) [6] and Gated Recurrent Unit (GRU) [7] are integrated into RNNs to help capture more reliable inter-frame temporal dependency [1, 8, 9, 10, 11, 12, 13, 14, 15]. In general, to save the computation resources and help predictive units to better perceive visual information, the fully connected layers in the predictive memories are replaced by convolutional layers in the above methods. Although the spatial receptive field of the unit has been improved by the integrated convolutional layers, the temporal receptive field is still narrow, and it is difficult for the unit at current time step to perceive what has happened in a longer past, which severely restrict the model expressivity to inter-frame motion information and the performance in predicting videos with complex scenarios and high resolutions is far from satisfactory.

Some works have attempted to broaden the temporal receptive field for the predictive units using 3D convolutional layers [16, 17]. However, the temporal receptive field is mainly determined by the kernel size of the integrated convolutional operators and the temporal dimension still needs to be set to a small value to meet the computation load requirement. Since then, the explorations for broadening the temporal receptive field for predictive models have been shelved and a variety of works begin to explore other ways to improve the expressivity of the model on videos with complex scenarios, which can be roughly categorized into two types, the structure-oriented methods and the loss-oriented methods. The structure-oriented methods utilized deep stochastic models to predict different futures for different samples based on their latent variables [18, 19, 20, 21]. However, the computation load of these methods is typically high, preventing their practicability in real world. And the loss-oriented methods aim to improve the traditional mean square error (MSE) based loss functions to generate more naturalistic results. Generative adversarial networks (GANs) [22, 23, 24, 25], perceptual loss [26] and so on have been utilized to generate results with higher perceptual quality. In spite of the explorations made by the above methods, only limited model performance improvements have been achieved and the unsatisfactory temporal receptive field still restricts the model performance in capturing reliable motion information between frames.

To solve the above problem, we propose the Motion-Aware Unit (MAU) to improve the model expressivity in capturing motion information by efficiently broadening the temporal receptive field. In particular, for each MAU, two modules are designed, the attention module and the fusion module. The attention module is designed for efficient attention and the fusion module is designed for efficient fusion. In particular, the attention module aims to help the unit to pay different levels of attention to the temporal states in the broadened temporal receptive field based on the corresponding spatial correlation scores. Using the attention scores, the temporal states can be aggregated to a more reliable augmented motion information (AMI) with a low computation load. The fusion module aims to further aggregate the augmented motion information (AMI) and the appearance information (the spatial state from current time step) using only two update gates. Moreover, an information recalling scheme is applied to further preserve the visual details of the predictions. Experimental results show that the proposed MAU can outperform other state-of-the-art methods on both video prediction and early action recognition tasks.

# 2 Related Work

In this section, we introduce the learning-based video prediction methods in detail. Due to the unique power in modeling sequential data, RNNs are first utilized to model videos. Ranzato *et al.* [5] utilized RNNs to propose a baseline model for unsupervised feature learning on video data. Srivastava *et al.*

[8] further integrated Long Short-Term Memories (LSTMs) [6] into RNN-based models to capture the temporal long-short term dependencies for videos, which is denoted as FC-LSTM. However, the fully connected layers in FC-LSTM are computation-expensive, which limits its practicability in real world. To further reduce the computation load and increase local perceptions to visual data for LSTMs, Shi *et al.* [9] proposed the ConvLSTM by replacing the fully connected layers with convolutional layers. Ballas *et al.* [10] further integrated convolutional layers into Recurrent Gated Unit (GRU) [7], denoted as ConvGRU, which has achieved a similar performance but with lower computation load compared with ConvLSTM. Motivated by the success achieved from the above video prediction methods, RNNs have been further explored to predict videos with higher visual quality and video prediction models are beginning to be applied into more research areas, such as robotics [1], precipitation nowcasting [11] and so on. However, the above works merely focus on exploring the temporal dependency but ignore the spatial features for videos. Wand *et al.* proposed PredRNN [13] to solve this problem by adding a spatial information processing module for ConvLSTMs. And to solve the gradient propagation difficulties in PredRNNs, Wang *et al.* further designed a Gradient Highway Unit [14] and the new model was denoted as PredRNN++.

Although the above RNN-based models have achieved some satisfactory results, the datasets utilized are either with simple scenarios or low resolutions and the model performance on videos with complex scenarios and high resolutions is still far from satisfactory. One main reason for this is that the predicted frames at current time step still mainly depend on the inputs from current time step and the temporal receptive field of the predictive model is narrow, which is not enough to predict reliable motion information for the next time step. To solve this problem, Wang *et al.* employed 3D convolutional layers into PredRNNs and utilized multi-term temporal states to help increase temporal receptive field for the predictive unit, which is denoted as E3D-LSTM [16]. However, the computation load of E3D-LSTM is extremely high due to the integrated 3D convolutional operators and the recall gate. To further improve the model expressivity for the predictive models, many other methods have also been proposed, which can be summarized into four types. The first type aims to protect the visual details for the predictions [12, 17, 27, 28]. The second type aims to predict different futures rather than an averaged future for each sample based on their latent variables [18, 19, 21]. The third type aims to refer to deeper models to increase the model expressivity [20, 29, 30]. And the last type aims to improve the loss functions to generate more naturalistic results [22, 23, 24, 25, 26]. However, in the above methods, the temporal receptive field problem was still not fully explored. In this paper, we propose the motion-aware unit (MAU), which can efficiently broaden the temporal receptive field to predict more reliable motion information for videos and the computation load is relatively low. In addition, MAU can be easily employed in other predictive models.

## 3 Method

### 3.1 Problem formulation

A video prediction model typically takes a video clip $\{v_1, ..., v_i\}$ as the inputs and outputs the future video clip $\{\hat{v}_{i+1}, ..., \hat{v}_T\}$. And we want to optimize the following problem,

$$\min \sum_{t=i+1}^{T} [\mathcal{D}(\hat{v}_t, v_t)], \qquad (1)$$

where $\hat{v}_t$ denotes the predicted frame at time step $t$, $\mathcal{D}$ denotes the loss function, such as the $\mathcal{L}_1$, $\mathcal{L}_2$ loss functions and so on.

To optimize the above problem, the error between the predicted frame $\hat{v}_t$ and the ground truth $v_t$ is expected to be as small as possible. For most of the RNN-based video prediction methods, the frames are usually progressively predicted where the proposed model predicts one frame for each time step. In particular, the predicted frame $\hat{v}_t$ depends on the spatial information, i.e. the frame input at the previous time step $v_{t-1}$ and the temporal information, i.e. the transited motion information $T_{t-1}$. However, as the time steps move on, the predicted error in Eq. 1 dramatically accelerates due to the increasing uncertainty of the transited temporal information $T_{t-1}$. To solve this problem, the predictive unit needs to search for more useful temporal information from a wider range of time steps, i.e. the temporal receptive field is needed to be broadened. For time step $t$, the temporal receptive field is probably to be set as $t - 1$. However, it may be not efficient and necessary to utilize all the previous frames, thus the receptive field is more likely to be set to a fixed value $\tau$. Although recent

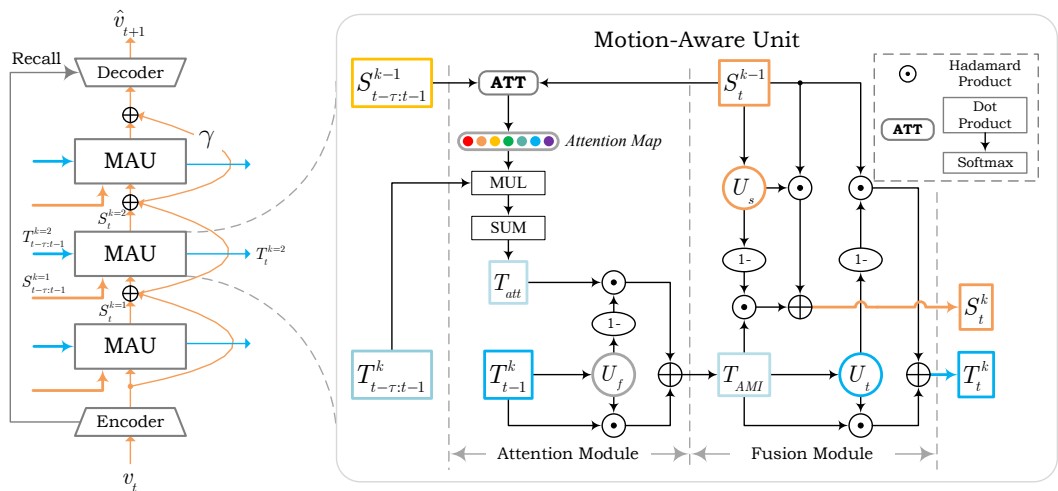

Figure 1: Left: The structure of the predictive network with stacked MAUs. Right: The structure of the proposed motion-aware unit: MAU. $T_t^k$ denotes the temporal state at time step $t$ from layer $k$. $S_t^k$ denotes the spatial state at time step $t$ from layer $k$. **SUM** and **MUL** represent summation and multiplication.

work E3D-LSTM [16] has attempted to address this problem and the temporal receptive field is broadened, the computation load is extremely high with only a limited performance improvement (mainly achieved by the integrated 3D convolutional layers), which indicates the broadened temporal receptive field may not been fully used.

To ensure that the broadened temporal receptive field can be fully utilized, two problems are needed to be solved,

- The temporal states in current receptive field should be aggregated according to their importance.

- The motion information from the aggregated temporal state and the appearance information from the spatial state should be fused reasonably.

To solve the above two problems, we propose the Motion-Aware Unit (MAU) and the predicted frame $\hat{v}_t$ can be represented as follows,

$$\hat{v}_t = Dec[MAU(Enc(v_{t-\tau:t-1}))], \tag{2}$$

where $Enc(\cdot)$ denotes the encoder which is utilized to extract deep features from the video input and $Dec(\cdot)$ denotes the decoder which is utilized to map the predicted features to the frames. $MAU(\cdot)$ denotes the proposed motion-aware unit, which will be detailedly introduced in Section 3.2.

### 3.2  The proposed Motion-Aware Unit: MAU

In this section, we introduce the structure details of MAU, as shown in Fig. 1 (right). Each MAU consists of two modules: the attention module and the fusion module. Typically, to improve the model expressivity, multiple MAUs are stacked, as shown in Fig. 1 (left). In particular, at time step $t$ in layer $k$, two inputs will be fed into the MAU: the temporal states set $T_{t-\tau:t-1}^k$ from previous $\tau$ time steps and the spatial states set $S_{t-\tau:t}^{k-1}$ from previous $\tau + 1$ time steps.

To solve the first problem in Section 3.1, we design the attention module, which aims to help the predictive unit to pay different levels of attention to different historical temporal states. The expected situation is that the predictive can always pay the highest level of attention to the most correlated states. Now the problem comes to how to quantify the correlations between different temporal states. Considering the fact that the visual quality of each frame can be the most factors to evaluate a video prediction model, the correlations between the corresponding spatial states can be an optimal choice.

Based on the analysis, the attention score $\alpha_j$ for the temporal state $T_{t-j}^k$ can be denoted as follows,

$$\alpha_j = \frac{e^{q_j}}{\sum_{i=1}^{\tau} e^{q_i}},$$
$$q_i = SUM(S_{t-i}^{k-1} \odot S'), \quad i = 1, ..., \tau,$$
$$S' = W_s * S_t^{k-1}, \tag{3}$$

where $\odot, *$ denote the Hadamard product and the convolutional operator, respectively. Using the computed attention score, the temporal state set can be aggregated as follows,

$$T_{att} = \sum_{j=1}^{\tau} \alpha_j \cdot T_{t-j}^k. \tag{4}$$

$T_{att}$ can be treated as the long-term motion information. However, besides the long-term motion information $T_{att}$, the short-term motion information $T_{t-1}^k$ is also needed to be utilized to strengthen the final motion information. We define a fusion gate $U_f$ to control the fusion process, which can be shown as follows,

$$U_f = \sigma(W_f * T_{t-1}^k),$$
$$T_{AMI} = U_f \odot T_{t-1}^k + (1 - U_f) \odot T_{att}, \tag{5}$$

where $\sigma$ denotes the sigmoid function, $T_{AMI}$ denotes the augmented motion information.

To solve the second problem in Section 3.1, we design the fusion module to aggregate the motion information in the augmented motion information $T_{AMI}$ with the appearance information in the current input $S_t^{k-1}$. To control the fusion ratios for both temporal and spatial information, two update gates are denoted as follows,

$$U_t = \sigma(W_{tu} * T_{AMI}),$$
$$U_s = \sigma(W_{su} * S_t^{k-1}), \tag{6}$$

where $U_t$ denotes the temporal update gate and $U_s$ denotes the spatial update gate. Using both gates, the aggregating process can be conducted as follows,

$$T_t^k = U_t \odot (W_{tt} * T_{AMI}) + (1 - U_t) \odot (W_{st} * S_t^{k-1}),$$
$$S_t^k = U_s \odot (W_{ss} * S_t^{k-1}) + (1 - U_s) \odot (W_{ts} * T_{AMI}) + \gamma \cdot S_t^{k-1}, \tag{7}$$

where the residual-like term $\gamma \cdot S_t^{k-1}$ is utilized to stabilize the training process.

In particular, as shown in Fig. 1 (left), the frame input $v_t$ is encoded to deep features by the encoder and the predicted spatial state is decoded back to the frame by the decoder, represented as follows,

$$S_0^t = Enc(v_t),$$
$$\hat{v}_{t+1} = Dec(S_t^N), \tag{8}$$

where $N$ denotes the total number of the employed MAUs.

### 3.3 Information recalling scheme

Considering the information loss problem during encoding, an information recalling scheme between encoders and decoders have been employed, which is defined as the information recalling scheme and can be represented as follows,

$$D_l = Dec_l(D_{l-1} + E_{-l}), \quad l = 1, ..., N, \tag{9}$$

where $D_l, E_{-l}$ denote decoded features from the $l^{th}$ layer of the decoder and the encoded features from the $l^{th}$ from the last layer of the encoder. $Dec_l$ denotes the $l^{th}$ layer of the decoder. On the basis of the above information recalling scheme, the decoders can recall multi-level encoded information back and the visual quality of the predictions is better.

Figure 2: The qualitative results from different methods on the Moving MNIST dataset.

## 4 Experiments

### 4.1 Implementations

We evaluate the proposed MAU on five datasets, the Moving MNIST dataset [8], the KITTI dataset [31], the Caltech Pedestrian dataset [32], the TownCentreXVID dataset [33] and the Something-Something V2 dataset [34]. The number of the hidden state channels of MAUs are set to 64 and the integrated convolutional operators are set with a kernel size $5 \times 5$ and stride 1. All experiments are optimized with the Adam optimizer. To stabilize the training process, we employ layer normalization operators after each integrated convolutional layer in MAUs.

### 4.2 Video Prediction

We conduct video prediction experiments on the the Moving MNIST dataset, the KITTI dataset, the Caltech Pedestrian dataset, and the TownCentreXVID dataset. The detailed experimental settings are summarized in Table 1. All models are optimized with the MSE loss function.

Table 1: Experimental settings. **MAUs** denotes the number of the stacked MAUs. **Train** and **Test** denotes the number of frames as the inputs and the outputs while training and testing.

| Dataset | Resolution | MAUs | Hidden_channels | Kernel | Train | Test | $\tau$ | $\gamma$ |
|---|---|---|---|---|---|---|---|---|
| Moving MNIST | $1 \times 64 \times 64$ | 4 | 64 | $5 \times 5$ | $10 \to 10$ | $10 \to 10$ | 5 | 0.0 |
| KITTI & Caltech | $3 \times 128 \times 160$ | 16 | 64 | $5 \times 5$ | $10 \to 1$ | $10 \to 10$ | 5 | 1.0 |
| TownCentreXVID | $3 \times 1088 \times 1920$ | 16 | 64 | $5 \times 5$ | $4 \to 1$ | $4 \to 4$ | 5 | 1.0 |

#### 4.2.1 Moving MNIST

The Moving MNIST dataset can be the most widely-used dataset in video prediction, where each sequence consists of 20 successive frames with 2 digits randomly placed. Each frame is with a size of $64 \times 64$. In our experiments, sequences are generated from the training set of the standard MNIST dataset [35] and we utilize the test set collected by Srivastava *et al.* [8] to evaluate the proposed model. The Mean Square Error (MSE), the Structural Similarity Index (SSIM) are employed to indicate the visual quality of the predictions.

Fig. 2 shows the visual results predicted from different methods, where the proposed MAU significantly outperforms other methods, especially for the prediction in the last two time steps. Table 2 summarizes the quantitative scores of the predictions from different methods. The proposed MAU has achieved the best scores compared with other state-of-the-art methods. And the employed recalling scheme can help further improve the model performance.

In addition, the parameters and inference time are also recorded in Table 3. For a fair comparison, all models are implemented with the same encoder and decoder with the same number of predictive units. In particular, for the proposed MAU, the recalling scheme is disabled and all models are trained

Table 2: Quantitative results of different methods on the Moving MNIST dataset (10 frames → 10 frames). Lower MSE and higher SSIM scores indicate better visual quality. The results of the compared methods are reported in [36].

| Method | Moving MNIST | |
| | SSIM/frame↑ | MSE/frame↓ |
| --- | --- | --- |
| ConvLSTM (NeurIPS2015) [9] | 0.707 | 103.3 |
| FRNN (ECCV2018) [12] | 0.819 | 68.4 |
| VPN (ICML2017) [29] | 0.870 | 70.0 |
| PredRNN (NeurIPS2017) [13] | 0.869 | 56.8 |
| PredRNN++ (ICML2018) [14] | 0.898 | 46.5 |
| MIM (CVPR2019) [15] | 0.910 | 44.2 |
| E3D-LSTM (ICLR2019) [16] | 0.910 | 41.3 |
| CrevNet (ICLR2020) [17] | 0.928 | 38.5 |
| MAU (w/o recalling) | 0.931 | 29.5 |
| MAU | **0.937** | **27.6** |

Table 3: Ablation study on the Moving MNIST dataset (10 frames → 10 frames). For fair comparison, the encoders and decoders are with the same structure for all models and All models are trained using Adam optimizer based on the MSE loss.

| Method | Backbone | MSE↓ | SSIM↑ | Parameters | Inference time |
| --- | --- | --- | --- | --- | --- |
| ConvLSTM (NeurIPS2015) [9] | 4×ConvLSTMs | 102.1 | 0.747 | 0.98M | **16.47s** |
| ST-LSTM (NeurIPS2017) [13] | 4×ST-LSTMs | 54.5 | 0.839 | 1.57M | 17.74s |
| Casual-LSTM (ICML2018) [14] | 4×Casual-LSTMs | 46.3 | 0.899 | 1.80M | 21.25s |
| MIM (CVPR2019) [15] | 4×MIMs | 44.1 | 0.910 | 3.03M | 45.13s |
| E3D-LSTM (ICLR2019) [16] | 4×E3D-LSTMs | 40.1 | 0.912 | 4.70M | 57.21s |
| RPM (ICLR2020) [17] | 4×RPMs | 42.0 | 0.922 | 1.77M | 18.01s |
| MotionGRU (CVPR2021) [28] | 4×MotionGRUs | 34.3 | 0.928 | 1.16M | 17.58s |
| MAU | 4×MAUs | **29.5** | **0.931** | **0.78M** | 17.34s |

Table 4: Model performance of MAU with different temporal receptive field $\tau$. In particular, $\gamma = 0, \lambda = 0$. The percentage values are calculated based the MAU with $\tau = 1$.

| | $\tau = 1$ | $\tau = 3$ | $\tau = 5$ | $\tau = 10$ |
| --- | --- | --- | --- | --- |
| MSE/frame | 33.4 | 32.2 (↓3.6%) | 29.5 (↓11.7%) | 29.3 (↓12.3%) |
| Inference time | 14.90s | 15.85s (↑6.4%) | 17.36s (↑16.5%) | 20.23s (↑35.8%) |

using Adam optimizer based on the MSE loss. We record the parameters for a single unit and the inference time is recorded over 800 samples. The summarized results show that the MAU can achieve the best scores with the fewest parameters and a relatively low computation load.

Table 4 shows the model performance with different temporal receptive field $\tau$ (w/o recalling). Although the quality of the predictions becomes better as the temporal receptive field increases, the computation load also dynamically increases. Thus, $\tau$ is typically set to an appropriate value to achieve a satisfactory trade-off between the visual quality and the computation load. In particular, we set $\tau = 5$ in this paper.

### 4.2.2 KITTI and Caltech Pedestrian datasets

We use two car-mounted camera video datasets to evaluate the performance of MAU in real scenarios. KITTI and Caltech Pedestrian datasets are very similar to the real-world scenarios, which are collected to train the autonomous vehicles.

We follow the experimental settings in [37], where all frames are cropped and resized to $128 \times 160$. The proposed model is trained on the KITTI dataset and tested on the Caltech Pedestrian dataset. In

particular, the frame rate of the Caltech Pedestrian dataset is adjusted to the same as KITTI (10 fps). A total of 32373 sequences are for training and 7725 sequences for testing. In addition, the proposed model is trained to predict the next frame with the first 10 frames as the inputs. While testing, the temporal period of the predictions is extended to 10 frames.

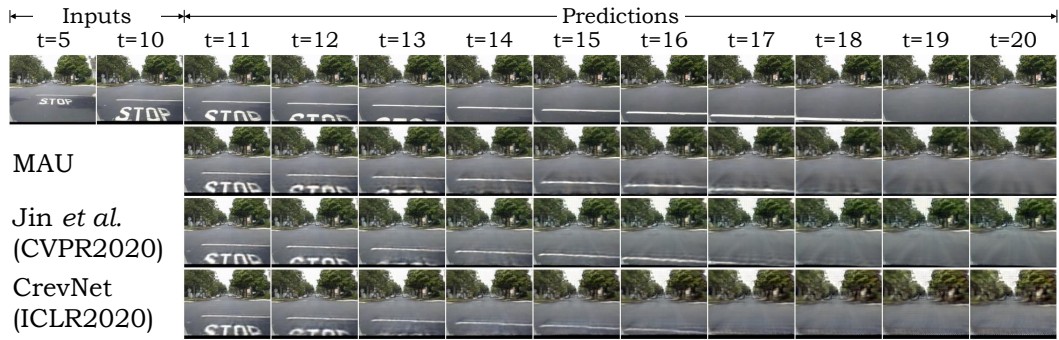

Figure 3: The qualitative results from different methods on the Caltech Pedestrian dataset.

Table 5: Quantitative results of different methods on the Caltech Pedestrian dataset. Lower MSE, LPIPS scores and higher SSIM, PSNR scores indicate better frame-level visual quality (10 frames → 1 frame). Lower FVD score indicates better sequence-level visual quality (10 frames → 10 frames). The results of the compared methods are partially reported in [36].

| Method | Caltech Pedestrian | | | | |
| | MSE($10^{-3}$)↓ | SSIM↑ | PSNR↑ | LPIPS($10^{-2}$)↓ | FVD/10 frames↓ |
| --- | --- | --- | --- | --- | --- |
| BeyondMSE (ICLR2016) [26] | 3.42 | 0.847 | - | - | - |
| MCnet (ICLR2017) [38] | 2.50 | 0.879 | - | - | - |
| CtrlGen (CVPR2018) [39] | - | 0.900 | 26.5 | - | - |
| PredNet (ICLR2017) [37] | 2.42 | 0.905 | 27.6 | 9.89 | 2860.8 |
| ContextVP (ECCV2018) [40] | 1.94 | 0.921 | 28.7 | 9.53 | 2451.6 |
| E3D-LSTM (ICLR2019) [16] | 2.12 | 0.914 | 28.1 | 10.02 | 2311.2 |
| Kwon et al. (CVPR2019) [24] | 1.61 | 0.919 | 29.2 | 8.03 | 1663.2 |
| CrevNet (ICLR2020) [17] | 1.55 | 0.925 | 29.3 | 9.11 | 1709.6 |
| Jin et al. (CVPR2020)[27] | 1.59 | 0.927 | 29.1 | 8.99 | 1441.1 |
| MAU (w/o recalling) | 1.34 | 0.939 | 29.4 | 8.51 | 1269.9 |
| MAU | **1.24** | **0.943** | **30.1** | **8.04** | **1204.0** |

Fig. 3 shows the generated examples from the proposed MAU and two latest state-of-the-art methods. From the visual results, the proposed MAU can predict more clear traffic signs ($t = 16$), which indicates the MAU can capture more reliable temporal dynamics for videos. Table 5 shows the quantitative results from different methods, where the proposed MAU outperforms other methods in all scores. In particular, the recalling scheme can further improve the model performance.

### 4.2.3 TownCentreXVID

In this section, we evaluate the proposed model on a surveillance dataset, TownCentreXVID, which is more close to the real scenarios and with a resolution of $1920 \times 1080$. TownCentreXVID dataset contains a total of 7500 frames. To further evaluate the model performance in predicting high-resolution videos, all frames are not resized. The first 4500 frames are for training and the last 3500 frames are for testing. Fig. 4 shows the visual results from different methods, where the proposed method can generate much better visual details compared with other state-of-the-art methods.

We further conduct object detection tasks on the predictions from different methods using the pre-trained Yolov5s model [41] and the results are shown in Fig. 5. More persons have been detected by the pre-trained Yolo model from the predictions generated from MAU and the confidence is higher than others.

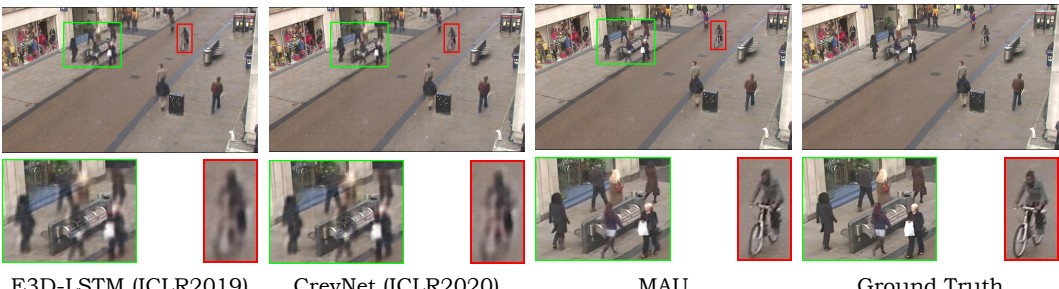

| E3D-LSTM (ICLR2019) | CrevNet (ICLR2020) | MAU | Ground Truth |

Figure 4: Qualitative results from different methods on the TownCentreXVID dataset (4 frames → 1 frame).

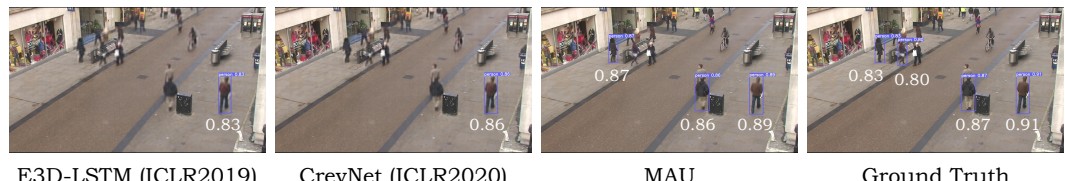

| E3D-LSTM (ICLR2019) | CrevNet (ICLR2020) | MAU | Ground Truth |

Figure 5: Object detection experiments on the predictions (4 frames → 1 frame) from different methods using Yolov5s pre-trained model [41]. Confidence threshold is set to 0.8.

Table 6: Quantitative results of different methods on the TownCentreXVID dataset (4 frames → 4 frames). Higher SSIM and PSNR scores indicate better objective quality. Lower LPIPS score indicates better perceptual quality.

| Method | TownCentreXVID | | | | | |
| | $t = 5$ | | | $t = 8$ | | |
| | PSNR↑ | SSIM↑ | LPIPS($10^{-2}$)↓ | PSNR↑ | SSIM↑ | LPIPS($10^{-2}$)↓ |
|---|---|---|---|---|---|---|
| ConvLSTM (NeurIPS2015) [9] | 27.22 | 0.894 | 39.90 | 23.29 | 0.876 | 46.12 |
| PredRNN (NeurIPS2017) [13] | 28.95 | 0.921 | 32.48 | 23.82 | 0.885 | 37.85 |
| PredRNN++ (ICML2018) [14] | 29.50 | 0.926 | 30.59 | 24.37 | 0.894 | 39.54 |
| E3D-LSTM (ICLR2019) [16] | 29.70 | 0.929 | 29.47 | 24.34 | 0.901 | 36.82 |
| CrevNet (ICLR2020) [17] | 30.12 | 0.933 | 27.87 | 24.62 | 0.910 | 33.70 |
| MAU (w/o recalling) | 30.61 | 0.937 | 25.87 | 25.52 | 0.913 | 32.42 |
| MAU | **31.87** | **0.969** | **8.28** | **27.14** | **0.942** | **12.89** |

In addition, Table 6 summarized the detailed quantitative results from MAU and other methods, where MAU has achieved the best objective (PSNR, SSIM) and perceptual (LPIPS) scores.

### 4.3 Early action recognition: Something-Something V2

The Something-SomethingV2 dataset is a large collection of labeled video clips that show humans performing pre-defined basic actions with everyday objects. The whole dataset consists of 174 categories of videos. The training set contains 168,913 videos and the validation set consists of 24,777 videos. The resolution of each video is $240 \times 427$. To further evaluate the performance in modeling high-level spatiotemporal representations for videos, the early action recognition task is conducted, which aims to categorize the whole video after observing only the front part of the videos. To accurately predict an activity category for current video, models need to extract useful spatiotemporal representations from merely the front part of the whole frames to predict a reliable future. In particular, we utilize the front 25% and 50% frames of each video to conduct this task, respectively. For each video, a total of 20 frames are sampled which can cover the whole temporal period. Each frame is resized to $128 \times 128$. For 25% early action recognition task, 5 frames are as

the inputs to predict the next 15 frames. For 50% early action recognition task, 10 frames are as the inputs to predict the next 10 frames.

All models are first pre-trained to perform video prediction task with the front 25% or 50% frames as the inputs to predict the remaining 75% or 50% frames. After convergence, a total of 19 hidden states extracted from the pre-trained models are concatenated to train the classifier, which consists of 3 3D convolutional layers. In particular, the concatenated hidden states are transformed from $C \times T \times H \times W : 64 \times 19 \times 16 \times 16$ to $174 \times 1$. To evaluate the performance of different predictive units in modeling spatiotemporal representations for videos, we utilize various predictive units as the backbone of the early action recognition model. For fair comparison, the encoder, decoder and classifier are set with the same structure for all models. A total of 16 predictive units are stacked for all models. The experimental results are summarized in Table 7. On the one hand, the predictive model with MAU has achieved the best PSNR score of the predictions. On the other hand, based on the learned spatiotemporal representations from the pre-trained models, the proposed MAU can also obtain the highest Top1 and Top5 accuracies.

Table 7: The results of the early action recognition experiment of different methods on the Something-Something V2 dataset.

| Method | Something-SomethingV2 | | | | | |
| | Front 25% | | | Front 50% | | |
| | PSNR↑ | top-1↑ | top-5↑ | PSNR↑ | top-1↑ | top-5↑ |
|---|---|---|---|---|---|---|
| ST-LSTM (NeurIPS2017) [13] | 14.87 | 3.77 | 14.17 | 15.78 | 8.91 | 19.18 |
| Casual-LSTM (ICML2018) [14] | 15.34 | 4.14 | 14.67 | 16.51 | 9.57 | 22.57 |
| E3D-LSTM (ICLR2019) [16] | 16.21 | 4.76 | 14.98 | 17.05 | 9.85 | 24.24 |
| RPM (ICLR2020) [17] | 16.53 | 4.98 | 15.07 | 17.57 | 10.01 | 24.51 |
| MotionGRU (CVPR2021) [28] | 17.01 | 5.11 | 15.16 | 17.86 | 10.22 | 27.65 |
| MAU | **17.36** | **5.40** | **19.00** | **18.47** | **10.60** | **30.90** |

## 5   Conclusion

We proposed the Motion-aware Unit (MAU) for video prediction and beyond. The motion-aware unit can take advantage in the broadened temporal receptive field, where more temporal states can be simultaneously perceived. In particular, the proposed unit are constructed based on the attention mechanism, which consists of two modules, the attention module and the fusion module. In particular, the attention module can extract attention weights from multi-term spatial states for multi-term temporal states. And the augmented motion information (AMI) can be aggregated from the multi-term temporal states. Then the fusion module is utilized to further aggregate the AMI and the spatial information to predict the final video frame. Although more states have been aggregated, the computation load is still relatively low because of the efficient structure in MAU. Moreover, an information recalling scheme was employed to help the decoders recall more multi-level encoded information. The proposed model was evaluated on two tasks, the video prediction task and the video classification task. Experimental results showed the proposed MAU can outperform other state-of-the-art methods on both tasks.

## 6   Limitations and Future Work

There are lots of issues that urgently need to be solved in the video prediction area and this work is also far from completely solving all the problems, such as high-resolution video prediction, low accuracy in action recognition, not practical enough to be applied into other downstream tasks and so on. All these problems need to be carefully taken into consideration in our future work. In addition, videos are still predicted in a deterministic way rather than a stochastic way, which may not practical enough for the complex real-world scenarios. Thus, in our future work, we will explore more to combine the spatiotemporal dynamics in videos into stochastic models, which are more similar to real-world scenarios.

## Acknowledgments and Disclosure of Funding

This work was supported in part by the National Natural Science Foundation of China (62072008, 62025101), PKU-Baidu Fund (2019BD003) and High-performance Computing Platform of Peking University, which are gratefully acknowledged.

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
