# OpenReview forum: "MAU: A Motion-Aware Unit for Video Prediction and Beyond"
_NeurIPS.cc/2021/Conference — NeurIPS 2021 Poster_

### Official Review · Reviewer_k1Ye · 2021-07-12

**Rating:** 6
**Confidence:** 3

**Summary:**

This paper proposes a Motion-Aware Unit (MAU) for video prediction tasks. The MAU includes two modules, the attention module and the fusion module. The attention module aims to learn the correlation between the current spatial state and previous spatial states. The learnt attention map can be used for augmenting the motion information (AMI). The fusion module is used for aggregating the augmented motion information (AMI) and current spatial state. The authors have demonstrated superior performance compared to state-of-the-arts.

**Limitations And Societal Impact:**

The authors have not stated negative parts. Current video prediction tasks are still low resolution (max $128\times 128$), which might not be suitable for real-life applications. Future work can be extended based on high resolution.


**Main Review:**

Apart from the good results presented in the paper, I have some questions about the paper as below:
1) P2 L67, the authors mentioned the good performance on the early action recognition task, but I have not found early action related experiment presented in the paper.
2) P4 L147, the authors clarify that they aim to quantify the correlation between different temporal states. But for **2nd line in eq.3**, how the Hadamard Product operation between two spatial states is able to quantify the correlation between matrices?
3) Do SUM and MUL in Figure 1 represent summation and multiplication? Needs an explanation here in the caption and main text.
4)  **2nd and 3rd lines in eq.3**, I suppose the objective of the 2nd line is to calculate the correlation between one state $t$ and all previous temporal states in window $\tau$ based on this state. If so, the subscription of $q_i$ is confused, it should be $q_t$.
5) In the experiment part, are backbones in encoder and decoder the same as a backbone in previous approaches? This needs description in the experiment part e.g., what backbones are used in encoder and decoder as well as for other approaches.

**Time Spent Reviewing:**

3

---

> ### Author Response · Authors · 2021-08-08
> **We thank the reviewer for the valuable comments.**
>
> We thank all the valuable comments from the reviewer.
>
> $\textbf{Response to Comment 1:}$ The early action related experiments are presented in section 4.2.
>
> $\textbf{Response to Comment 2:}$ The correlation coefficient between two vectors $(x,y)$ can be typically defined as $R_{x,y} = \frac{cov(x,y)}{\sqrt{var(x)\cdot var(y)}}$, where $cov(x,y)$ denotes the covariance between $x$ and $y$. $var(\cdot)$ denotes the variance and $cov(x,y)=E[(x-E(x))\cdot(y-E(y))]$, where $E(\cdot)$ denotes the expectation. In the proposed model, we employ layer normalization after each convolutional operation, thus for each state $T$, we have $E(T)=0, var(T)=1$, based on which, $R_{x,y} = \frac{cov(x,y)}{\sqrt{var(x)var(y)}}=E(x\cdot y)$ where $E(x\cdot y)$ can be expressed as the summation of the hardmard between two states. The whole proof will be added in the revised paper.
>
>
> $\textbf{Response to Comment 3:}$ SUM and MUL in Figure 1 represent summation and multiplication. More explanations will be added in the caption of Figure 1 and the main text.
>
> $\textbf{Response to Comment 4:}$ Actually, $q_i$ is calculated for the $i^{th}$ previous spatial state $S_{t-i+1}^{k-1}$ rather than the current spatial state at time $t$. $i$ denotes the index of the previous spatial state in the temporal window. We will add more explanations in the revised paper to clarify this.
>
> $\textbf{Response to Comment 5:}$ All reported results in Table 2,5,6 are collected from the corresponding published paper and all qualitative results in Figure 2,3,4 are generated based on the official code. Considering the differences in the model structure of different methods, the encoder and the decoder are also different. We also notice this problem, thus in Table 3, we build a predictive model with different predictive memories integrated into. In this way, only the employed predictive memory can influence the model performance and the results can be more convincing.
>
> We didn't use the current published network as the backbone, such as VGG-net, resnet and so on. Both encoders and decoders are structured with the basic downing layer (DL) and the upsampling layer (UL). The parameters for the DL and UL are set as follows,
>
> DL: Conv(in\_channel, out\_channel, kernel\_size=(3,3), stride=(2,2), padding=(1,1)) $\rightarrow$ LeakyReLU(0.2)
>
> UL: Conv(in\_channel, out\_channel, kernel\_size=(3,3), stride=(2,2), padding=(1,1), output\_padding=(1,1)) $\rightarrow$ LeakyReLU(0.2)
>
> $\textbf{Moving MNIST:}$
>
> Inputs: $(1\times64\times64)$, channel, height, width
>
> Encoder: Conv(in\_channel=1, out\_channel=64, kernel\_size=1, stride=1, padding=0) $\rightarrow$ 2$\times$ DL(in\_channel=64, out\_channel=64)
>
> Internal features: $(64\times16\times16)$
>
> Decoder: 2$\times$UL(in\_channel=64, out\_channel=64) $\rightarrow$ Conv(in\_channel=64, out\_channel=1, kernel\_size=1, stride=1, padding=0)
>
> $\textbf{KITTI and Caltech Pedestrian:}$
>
> Inputs: $(3\times128\times160)$, channel, height, width
>
> Encoder: Conv(in\_channel=3, out\_channel=64, kernel\_size=1, stride=1, padding=0) $\rightarrow$ 3$\times$ DL(in\_channel=64, out\_channel=64)
>
> Internal features: $(64\times16\times20)$
>
> Decoder: 3$\times$UL(in\_channel=64, out\_channel=64) $\rightarrow$ Conv(in\_channel=64, out\_channel=1, kernel\_size=1, stride=1, padding=0)
>
> $\textbf{TownCentreXVID:}$
>
> Inputs: $(3\times1088\times1920)$, channel, height, width $\stackrel{reshape}{\longrightarrow}$ $(48\times272\times480)$
>
> Encoder: Conv(in\_channel=48, out\_channel=64, kernel\_size=1, stride=1, padding=0) $\rightarrow$ 4$\times$ DL(in\_channel=64, out\_channel=64)
>
> Internal features: $(64\times17\times30)$
>
> Decoder: 4$\times$UL(in\_channel=64, out\_channel=64) $\rightarrow$ Conv(in\_channel=64, out\_channel=1, kernel\_size=1, stride=1, padding=0)
>
> $\textbf{Something-SomethingV2:}$
>
> Inputs: $(3\times128\times128)$, channel, height, width
>
> Encoder: Conv(in\_channel=3, out\_channel=64, kernel\_size=1, stride=1, padding=0) $\rightarrow$ 2$\times$ DL(in\_channel=64, out\_channel=64)
>
> Internal features: $(64\times16\times16)$
>
> Decoder: 2$\times$UL(in\_channel=64, out\_channel=64) $\rightarrow$ Conv(in\_channel=64, out\_channel=1, kernel\_size=1, stride=1, padding=0)
>
> More technical details will be added in the revised paper.
>
> $\textbf{Response to Comment 6:}$ We will carefully summarize a limitation part in the revised paper.
>
> We hope the responses can address the concerns of the reviewer.

---

> > ### Comment · Reviewer_k1Ye · 2021-08-18
> > **Response to authors' feedback**
> >
> > Thanks for the authors' feedback. The authors' feedbacks have solved my questions. I will keep my initial score unchanged.

---

### Official Review · Reviewer_Bgyd · 2021-07-12

**Rating:** 6
**Confidence:** 4

**Summary:**

In this work, the authors introduce a Motion-Aware Unit (MAU) for video prediction. Basically, they discover the previous motion and aggregate it via attention module and fusion module. In the attention module, they compute correlation between spatial states and use it as attention for aggregation the previous temporal states. In the fusion module, they design spatial and temporal update gates for fusing attentive temporal state and current spatial state.

**Limitations And Societal Impact:**

Please refer to Main Review for the detailed comments.
1 Novelty is limited. The design is not quite new, based on the fact that attention for motion learning has been widely used in video understanding.
2 By the way, temporal shift module [TSM: Temporal Shift Module for Efficient Video Understanding, ICCV2019] is a popular mechanism for early action recognition. It would be interesting to see how it works in Table 7.



**Main Review:**

1. Originality: The novelty is relatively limited. Attentively discovering historical motion has been widely investigated in online action recognition [Skeleton-Based Online Action Prediction Using Scale Selection Network, TPAMI2020] and detection [Learning to Discriminate Information for Online Action Detection, CVPR2020]
2. Quality: The motivation, model design and experiments basically make sense.
3. Clarity: It is an OK paper for me.
4. Significance: It is probably useful for online action recognition and detection, etc.

**Time Spent Reviewing:**

1.5

---

> ### Author Response · Authors · 2021-08-06
> **All comments are very valuable for us.**
>
> We thank the reviewer for the valuable comments. Although attention for motion learning has been widely used in action recognition, it is new in the video prediction area. In addition, this work is not proposed to solve the early action recognition tasks. We only employ the early action recognition tasks to further evaluate the model expressivity to spatiotemporal dynamics. However, in our further work, we will carefully adopt this advice and will compare the proposed work with other action recognition methods (not only limited to early action recognition). Moreover, we will add the performance score of TSM (ICCV2019) in Table 7. We really thank all the valuable comments and hope the responses can address the concerns of the reviewer.

---

### Official Review · Reviewer_tpAZ · 2021-07-14

**Rating:** 7
**Confidence:** 5

**Summary:**

This paper proposes a method for video prediction that relies on a newly proposed Motion-Aware Unit (MAU). MAU acts as a temporal receptive field based on an attention mechanism that is used to aggregate features from previous frames in order to predict future frames. The aggregate features are combined with “content” features extracted from the last observed frame and the decoder is connected to the encoder via an “Information recalling scheme” that directly passes the last observed frame features from each layer of the encoder to each layer of the decoder. In experiments, the authors show superior performance against the baselines pixel based metrics and a perceptual metric.

**Limitations And Societal Impact:**

They have not addressed their limitations or societal impact of their work.

**Main Review:**

Strengths:
+ A Motion-Aware Unit (MAU) that enables direct retrieval of previous frames extracted features using attention.
+ Combination layers that aggregate motion and content information by choosing important information from each information stream.
+ Outperforms baselines

Weaknesses:

- Evaluation (Table5 and Table 6):
As far as I can tell, the experimental setup is described in Table 1. However, when I read the captions on top of Table 5 and 6, I see that the authors describe something different. They indicate that for the Caltech and TownCentreXVID experiments their test setups are (10->1) and (4->1), respectively. This is completely different to what Table1 says. The test setups in Table 1 are (10 -> 10) and (10->10). If we assume that they mean train setup, it is the same story. The train setups in Table 1 are (10->1) and (10->10). Can the authors clarify this? I would also suggest the authors include a sequence level evaluation metric such as Fretchet Video Distance. Frame level evaluations do not usually tell the full story video prediction since we need to also measure if the dynamics are also plausible. FVD repo: https://github.com/google-research/google-research/blob/master/frechet_video_distance/frechet_video_distance.py. Finally, the use of evaluation metrics are not consistent throughout the experiments. I understand that perceptual evaluations such as LPIPs may not make sense for the Bouncing MNIST experiments, but they are applicable to both Caltech and TownCentreXVID experiments. However, I only see perceptual evaluations on the latter. Can the authors explain why this is the case?

- Ablations:
The authors have provided ablations of \tau for the number of frames considered in the temporal receptive field. However, there are other pieces in the architecture/formulation that would be good to see how much they contribute to performance. For example, what happens in Equation 5 if we simply let T_{AMI} = T_{attn} or T_{AMI} = T^k_{t-1}? T_{attn}. Another place where an ablation is necessary is  Equation 7. What if we let S^k_t = W_{ts} * T_{AMI} + \gamma S^k-1_t? Or can we visualize U_s to see if the network simply decided to set U_s=0 during training? This would be informative to decide whether the current complexity is necessary or not.

- TownCentreXVID experiments:
Table 6 shows that the proposed method outperforms the baselines. However, it is very difficult to tell from Figure 4 whether the proposed method is simply copying the last observed frame or making an actual prediction of the future. From very closely inspecting MAU against the groundtruth, it looks like what MAU outputs is what seems to be the frame before the Grouthtruth. It has what seems to be a temporal progression. However, as I said before, it’s hard to tell without previous image context. In addition, the folder in the supplementary material where visual results should be for this dataset is empty. The author’s should have provided a baseline where the last observed frame is copied and evaluate it to check the numbers against the proposed method and make sure it’s not doing the same thing.

- Video results:
Papers on video prediction usually provide video results since it’s very difficult to tell the quality of the prediction from frames on a pdf. This would have clarified a lot of the concerns mentioned above.


- Limitations of this work?
The authors have not mentioned any limitations of this work, and also marked N/A in the limitations question of the checklist. Why are there no limitations? Did the proposed method completely solve the problem of video prediction? Does the method learn perfect features for the action recognition/anticipation task or other sequence downstream tasks? From reading the paper, I see there are a lot of limitations, some of which we should be able to see if we have videos to observe. Also the action recognition/anticipation experiments (quality of learned features verification) accuracy is really low. In the future, the authors should have a paragraph or two describing what is failing in their method and what needs to be done next so that other researchers can pick up from there or get inspired by the work. That’s what academia is all about.

Suggestions:
- Action recognition experiments against recognition methods:
The provided action recognition experiments are great to show that the proposed method learns something useful from predicting future frames. I also suggest the authors train a network to do action recognition from scratch and compare against it. It would give readers a good idea about the gap between self-supervised and fully-supervised feature learning for this task.

- Action recognition without “recalling” in the prediction training:
The “recalling” connections from encoder to decoder are basically skip connections which have been used by previous works to get better frame prediction and make learning easier. However, these skip connections could have a negative impact on feature learning since the network can just copy a large portion of the pixels in the input image. I wonder if the action recognition performance improves if the recalling connections are removed because the features have to represent the entire image during prediction.

Conclusion:
I really like the formulation of this method as it is simple and seems that it should work well. However, the experimental section has a lot of issues that I cannot look past because I like the method. Because of this, I am at the borderline/leaning towards rejection of this paper. I am looking forward to the author's response to hopefully address my doubts/concerns.


#####################################
##### Post rebuttal comments ############
#####################################

The authors addressed all my concerns. Therefore, I have decided to increase my score to accept.


**Time Spent Reviewing:**

4

---

> ### Author Response · Authors · 2021-08-08
> **All comments and suggestions have been carefully taken into consideration.**
>
> $\textbf{Response to Comment 1: Evaluation (Table5 and Table 6)}$ We thank the comments. There is something wrong with Table 1. The training and testing setups for the TownCentreXVID dataset have been revised to $(4\rightarrow1)$ and $(4\rightarrow1)$, respectively. In Table5, $(10\rightarrow1)$ indicates that only the performance scores at $t=11$ are reported. In our revised version, more performance scores at more time steps will be summarized. Moreover, Fig. 3 shows the qualitative results on the Caltech Pedestrian dataset with the testing setup as $(10\rightarrow10)$. The FVD evaluation metric has been employed in our revised paper. As shown in Table1, while testing, multiple frames will be generated on the Moving MNIST and Caltech datasets, and only one frame is predicted on the TownCentreXVID dataset due to the high computation load. Considering this situation, we only summarize the FVD score on the first two datasets, which are shown as follows,
>
> $\textbf{Moving MNIST}$
>
> ConvLSTM (NeurIPS2015): 152.97
>
> PredRNN (NeurIPS2017): 76.99
>
> PredRNN++ (ICML2018): 91.48
>
> E3D-LSTM (ICLR2019): 88.68
>
> CrevNet (ICLR2020): 63.57
>
> MAU w/o recalling: 39.74
>
> MAU: 36.96
>
> $\textbf{Caltech}$
>
> PredNet (ICLR2017): 2860.78
>
> ContextVP (ECCV2018): 2451.56
>
> E3D-LSTM (ICLR2019): 2311.22
>
> Kwon et al. (CVPR2019): 1663.21
>
> CrevNet (ICLR2020): 1709.56
>
> Jin et al. (CVPR2020): 1441.08
>
> MAU w/o recalling: 1269.91
>
> MAU: 1204.02
>
> Moreover, we further evaluate the performance of different methods on the Caltech Pedestrian dataset using LPIPS metric and the experimental results have been added in the revised paper, as shown in the followings $(\times 10^{-2})$,
>
> PredNet (ICLR2017): 7.47
>
> ContextVP (ECCV2018): 6.03
>
> E3D-LSTM (ICLR2019): 6.31
>
> Kwon et al. (CVPR2019): 4.91
>
> CrevNet (ICLR2020): 5.94
>
> Jin et al. (CVPR2020): 5.89
>
> MAU w/o recalling: 4.90
>
> MAU: 4.85
>
>
> We thank again for the valuable comments which have greatly helped us refine our work.
>
> $\textbf{Response to Comment 2: Ablations}$ The suggestions from the reviewer are great. More ablation studies will be conducted on Equation 5,7 to further explore see how much different terms contribute to the model performance (Recalling scheme is disabled). Some results on the Moving MNIST dataset are shown as follows,
>
>
> Moving MNIST: MSE/frame
>
> $T_{AMI}=T_{t-1}^k (U_f=1)$: 18.1
>
> $T_{AMI}=T_{att} (U_f=0)$: 16.4
>
> $S_t^k = W_{ss}\ast S_t^{k-1}+\gamma\cdot S_t^{k-1} (U_s=1)$: 26.0
>
> $S_t^k = W_{ts}\ast T_{AMI}+\gamma\cdot S_t^{k-1} (U_s=1)$: 21.1
>
> MAU: 9.7
>
>
> From the above results, the MAU obvious other methods which simply freeze the values in $U_f$ and $U_s$, indicating that the current complexity is definitely necessary.
> More ablation studies based on the above suggestions will be added in the revised paper.
> In our revised paper, all three gates will be visualized during training to help further prove this.  We thank the reviewer again for this valuable suggestion.
>
> $\textbf{Response to Comment 3: TownCentreXVID experiments}$ We thank the comments. The TownCentreXVID dataset is a survivance dataset with the background as the same for different time steps. Thus, the proposed model is more likely to only copy the background information from previous frames rather than the motion information. However, due to upload size limitations, we haven't contained all the datasets in the supplementary material. To solve this problem, we follow the reviewer's suggestion and propose a new baseline where the last observed frame is copied, and the results are summarized as follows,
>
> |Method|PSNR|SSIM|
>
> |Copying|30.01|0.929|
>
> |MAU|31.87|0.969|
>
> From the results above, the predictions from the proposed model are better than the copying frame.
> Moreover, in videos with high resolutions (1080p~4k), the background information is much more compared with the motion information, thus it is hard for the human eyes to percept the noticeable difference between the adjacent frames. However, from the above results, we can conclude the predictive model is really doing the predicting rather than copying. More qualitative results will be added in the revised paper to further highlight such slight visual differences.
>
> $\textbf{Response to Comment 4: Video results}$ We thank the comments. More video results will be provided in our project page after this work is published.
>
> $\textbf{Response to Comment 5: Limitations of this work}$ We thank the comments. We feel so sorry for not having constructed a limitation part into this paper. There are lots of issues that urgently needed to be solved in the video prediction area and this work is also far from completely solving all of the problems, such as high-resolution video prediction, low accuracy in action recognition, not practical enough to be applied into other downstream tasks and so on. All these problems need to be carefully taken into consideration. For this work and our future work, we will always add a limitation part to help researchers discuss and help push this area to a better and deeper place. We thank the reviewer again for the very valuable comments.
>
> $\textbf{Response to Suggestion 1: Action recognition experiments against recognition methods}$ We thank the valuable suggestions. In our future work, we plan to build a action recognition network with the proposed MAU as the backbone and compare the performance between current methods.
>
> $\textbf{Response to Suggestion 2: Action recognition without ``recalling'' in the prediction training}$ We thank the valuable suggestions. In fact, in this paper, the recalling scheme has been disabled for all the models in Table 7. Another experiment will be conducted with the recalling scheme enabled in the revised paper. In our future work, we will add a coefficient ($0\sim1$) multiply with the recalling information to explore the changing trend of the visual quality and the recognition accuracy.
>
> All suggestions have been adopted and we hope the responses can address the concerns of the reviewer.

---

> > ### Comment · Reviewer_tpAZ · 2021-08-31
> > **Response to rebuttal**
> >
> > I would like to thank the authors for their efforts in the rebuttal. The authors addressed almost all of my concerns. Therefore, I will increase my score towards acceptance. Having said that, I suggest the authors to try to provide multi-step prediction results on TownCentreXVID. I feel that this result could help readers see how much this work pushes the boundaries as this dataset is difficult to predict due to all the different types of objects present in the scene.

---

> > > ### Author Response · Authors · 2021-08-31
> > > **We thank the reviewer for the very valuable responses.**
> > >
> > > We thank the very valuable comments from the reviewer, and we hope the additional responses can address the remaining concerns of the reviewer.
> > >
> > > We have collected the multi-step prediction results on the $\textbf{TownCentreXVID}$ dataset (4 frames$\rightarrow$4frames, 1088$\times$1920) using almost all the GPU resources and the quantitative results are shown as follows (we didn't summarize the FVD score for that the minimum length of video inputs is 9 according to the source code of FVD),
> > >
> > > -----------------------------------------------------------------------------------
> > > Methods     $\quad\quad\quad\quad\quad\quad\quad\quad\quad\quad$$t=5$          $\quad\quad\quad\quad\quad\quad\quad$                             $t=8$
> > >
> > > $\quad\ \ \quad\quad\quad\quad\quad\quad\quad\quad\quad$PSNR$\quad$SSIM $\quad$LPIPS$\quad$             PSNR$\quad$SSIM$\quad$LPIPS
> > >
> > > CovLSTM    $\quad\quad\quad\quad\quad\quad$27.22 $\quad$0.894 $\quad$39.90             | $\quad$23.29 $\quad$0.876 $\quad$46.12
> > >
> > > PredRNN   $\quad\quad\quad\quad\quad\quad$28.95 $\quad$0.921 $\quad$32.48                 |   $\quad$23.82 $\quad$0.885 $\quad$37.85
> > >
> > > PredRNN++ $\quad\quad\quad\quad\quad$29.50 $\quad$0.926 $\quad$30.59              | $\quad$24.37 $\quad$0.894 $\quad$39.54
> > >
> > > E3D-LSTM $\ \ \quad\quad\quad\quad\quad$29.70 $\quad$0.929 $\quad$29.47                    |$\quad$24.34 $\quad$0.901 $\quad$36.82
> > >
> > > CrevNet $\quad\ \quad\quad\quad\quad\quad$30.12$\quad$ 0.933 $\quad$27.87                       |$\quad$24.62$\quad$ 0.910 $\quad$33.70
> > >
> > > MAU (w/o recalling) $\ \ \quad$30.84 $\quad$0.939 $\quad$24.07|$\quad$           25.52 $\quad$0.914 $\quad$30.87
> > >
> > > MAU $\ \ \quad\quad\quad\quad\quad\quad\quad$31.87 $\quad$0.969 $\quad$8.28                           |$\quad$ 27.14 $\quad$0.942 $\quad$12.89
> > >
> > > ------------------------------------------------------------------------------------------
> > >
> > > where higher PSNR, higher SSIM, and lower LPIPS scores indicate better results and the proposed method outperforms others. More qualitative results will be provided in our revised paper. In our future work, we will explore deeper into this area to further improve the model performance.
> > >
> > > We hope our responses can address the remaining concerns of the reviewer. We also hope the reviewer can increase the final score in the $\textbf{OpenReview system}$ after reading our responses which will inspire us a lot to do our future work. We really thank the comments from the reviewer.

---

> > > > ### Comment · Reviewer_tpAZ · 2021-08-31
> > > > **Very cool**
> > > >
> > > > Thank you very much for the "clutch" results. Good job!

---

### Official Review · Reviewer_LrcY · 2021-07-16

**Rating:** 6
**Confidence:** 4

**Summary:**

The paper proposes the Motion-Aware Unit (MAU) for video prediction. This module attempts to model the temporal information in videos by enlarging the temporal receptive field and aggregating it through an attention mechanism. The proposed module is employed in the bottleneck part of an autoencoder where the spatial and temporal information are combined at different MAU layers. The proposed method is compared to previous works on several benchmarks for next frames prediction and early action recognition showing state-of-the-art results.

**Limitations And Societal Impact:**

The authors do not highlight any limitations of the proposed method. Moreover, I do not find any potential negative societal impact regarding this work.

**Main Review:**

The proposed method shows interesting results on both the video prediction and early action recognition tasks. Although the idea of enlarging the temporal receptive field when dealing with sequence forecasting is not new [1], in the video prediction task, this idea is not already investigated. However, the method introduced is incremental as the main contribution stands on the aggregation of spatial and temporal features using an attention-based mechanism.

Figure 1 on the right is barely comprehensible; I suggest making it clearer. In line 110, the L1 and L2 terms are not introduced before; it would be better to state what they are clearly.

Many important details regarding the model and the experiments are missing.
- Which encoder and decoder backbones do you use for the experiments?
- Are the encoder and the decoder comparable with respect to previous works in the results reported in the Caltech Pedestrian, KITTI, and TownCentreXVID datasets?
- What is the dimensionality of the internal features of the model?
- Does such dimensionality change in the different benchmarks you used?
- How does such dimensionality impact the efficiency of the model?
- Is your architecture fully convolutional?

Lots of details for reproducing the results are also missing (training parameters, ...).

It would be useful to visualize the attention scores of the MAU module in order to better understand the behavior of the proposed model when it generates new frames.

References:

[1] Attention is All you Need, Vaswani et al, NIPS 2017.

**Time Spent Reviewing:**

6

---

> ### Author Response · Authors · 2021-08-08
> **All comments are very valuable to us and more experimental settings have been provided in the responses and the revised paper.**
>
> $\textbf{Response to Comment 1: Figure 1 and L1, L2 terms}$ We thank the valuable comments from the reviewer. For better understanding, we have added more explanations into the caption to make Figure1 right clearer. L1 term indicates the Mean Absolute Error (MAE), L2 term denotes the Mean Square Error (MSE). More detailed explanations have been provided in the revised paper.
>
> $\textbf{Response to Comment 2: Model details}$
> We thank the valuable comments.
>
> $\textbf{(1)}$ Encoder and decoder backbones. We didn't use the current published network as the backbone, such as VGG-net, resnet and so on. Both encoders and decoders are structured with the basic downsampling layer (DL) and the upsampling layer (UL). The parameters for the DL and UL are set as follows,
>
> DL: Conv(in\_channel, out\_channel, kernel\_size=(3,3), stride=(2,2), padding=(1,1)) $\rightarrow$ LeakyReLU(0.2)
>
> UL: Conv(in\_channel, out\_channel, kernel\_size=(3,3), stride=(2,2), padding=(1,1), output\_padding=(1,1)) $\rightarrow$ LeakyReLU(0.2)
>
> $\textbf{Moving MNIST:}$
>
> Inputs: $(1\times64\times64)$, channel, height, width
>
> Encoder: Conv(in\_channel=1, out\_channel=64, kernel\_size=1, stride=1, padding=0) $\rightarrow$ 2$\times$ DL(in\_channel=64, out\_channel=64)
>
> Internal features: $(64\times16\times16)$
>
> Decoder: 2$\times$UL(in\_channel=64, out\_channel=64) $\rightarrow$ Conv(in\_channel=64, out\_channel=1, kernel\_size=1, stride=1, padding=0)
>
> $\textbf{KITTI and Caltech Pedestrian:}$
>
> Inputs: $(3\times128\times160)$, channel, height, width
>
> Encoder: Conv(in\_channel=3, out\_channel=64, kernel\_size=1, stride=1, padding=0) $\rightarrow$ 3$\times$ DL(in\_channel=64, out\_channel=64)
>
> Internal features: $(64\times16\times20)$
>
> Decoder: 3$\times$UL(in\_channel=64, out\_channel=64) $\rightarrow$ Conv(in\_channel=64, out\_channel=1, kernel\_size=1, stride=1, padding=0)
>
> $\textbf{TownCentreXVID:}$
>
> Inputs: $(3\times1088\times1920)$, channel, height, width $\stackrel{reshape}{\longrightarrow}$ $(48\times272\times480)$
>
> Encoder: Conv(in\_channel=48, out\_channel=64, kernel\_size=1, stride=1, padding=0) $\rightarrow$ 4$\times$ DL(in\_channel=64, out\_channel=64)
>
> Internal features: $(64\times17\times30)$
>
> Decoder: 4$\times$UL(in\_channel=64, out\_channel=64) $\rightarrow$ Conv(in\_channel=64, out\_channel=1, kernel\_size=1, stride=1, padding=0)
>
> $\textbf{Something-SomethingV2:}$
>
> Inputs: $(3\times128\times128)$, channel, height, width
>
> Encoder: Conv(in\_channel=3, out\_channel=64, kernel\_size=1, stride=1, padding=0) $\rightarrow$ 2$\times$ DL(in\_channel=64, out\_channel=64)
>
> Internal features: $(64\times16\times16)$
>
> Decoder: 2$\times$UL(in\_channel=64, out\_channel=64) $\rightarrow$ Conv(in\_channel=64, out\_channel=1, kernel\_size=1, stride=1, padding=0)
>
> $\textbf{(2)}$ All reported results in Table 2,5,6 are collected from the corresponding published paper and all qualitative results in Figure 2,3,4 are generated based on the official code. Considering the differences in the model structure of different methods, the encoder and the decoder are also different. We also notice this problem, thus in Table 3, we build a predictive model with different predictive memories integrated into. In this way, only the employed predictive memory can influence the model performance and the results can be more convincing.
>
> $\textbf{(3)}$ The dimensionality of the internal features are summarized as follows:
>
> $\textbf{Moving MNIST:}$
> $(64\times16\times16)$, channel, height, width
>
> $\textbf{KITTI and Caltech Pedestrian:}$
> $(64\times16\times20)$
>
> $\textbf{TownCentreXVID:}$
> $(64\times17\times30)$
>
> $\textbf{Something-SomethingV2:}$
> $(64\times16\times16)$
>
> $\textbf{(4)}$ The dimensionality will change in the different benchmarks used in this paper. However, in all three datasets, the proposed MAU only uses the smallest dimensionality of the internal features to achieve the best performance.
>
> $\textbf{(5)}$ Such dimensionality will impact the efficiency of the model. The predictive units in the proposed model will consume most of the computation resources compared with other modules(encoder, decoder). Thus as the dimensionality of the internal features increases, the computation load will also dynamically increase, and the model performance will be also improved for the fact that more spatiotemporal information can be preserved. Thus, a trade-off between the computation load and the model performance needs to be explored. In the revised paper, we will further explore the correlations between such dimensionality and the model efficiency (computation load and performance). We thank this very valuable comment again and we will carefully adopt it in our work.
>
> $\textbf{(6)}$ The whole model is fully convolutional.
>
> More details about the experimental settings will be added in the revised paper and you can find more technical details from our code in the supplementary material.
>
> $\textbf{Response to Comment 3: Visualize the attention:}$ We thank the reviewer for this great comment and we will adopt this suggestion in the revised paper.
>
> $\textbf{Response to Comment 4: Limitations:}$ We are so sorry for not having a limitation part contained in this paper. In our revised paper, we will definitely highlight a limitation part.
>
> We really hope the responses can address the concerns of the reviewer.

---

### Decision · Program_Chairs · 2021-09-27

**Decision:**

Accept (Poster)

**Comment:**

There was a robust discussion between the reviewers on the merits of this work. The author response helped clarify many of the things that the reviewers asked for. All in all, I feel this work should be accepted at NeurIPS. While the reviewers as a whole felt that there is some lack of novelty in the work, and had some minor constructive feedback (e.g. more justification for not performing multi-step prediction in their TownCentreXVID experiments), the conclusion is that the contributions are sufficient and interesting enough to warrant acceptance.